# Research on Concrete Crack Detection in Hydropower Station Burial Engineering Based on Quantum Particle Technology

**DOI:** 10.3390/s25030683

**Published:** 2025-01-23

**Authors:** Yuanjiang Ma, Jun Fu, Qingsong Zhang, Xiaobing Liu, Bingxu Chen, Gang Yan, Hua Shi

**Affiliations:** 1Yingxiuwan Hydropower Plant, State Grid Sichuan Electric Power Company, Chengdu 611830, China; mayj1797@outlook.com (Y.M.); 13880012550@139.com (J.F.); 13628007310@163.com (Q.Z.); 2Key Laboratory of Fluid and Power Machinery, Ministry of Education, Xihua University, Chengdu 610039, China; liuxb@mail.xhu.edu.cn; 3Sichuan Key Laboratory of Fluid Machinery and Engineering, Xihua University, Chengdu 610039, China; 4School of Energy and Power Engineering, Xihua University, Chengdu 610039, China; 5Sichuan Sunlight Mozi Technology Co., Ltd., Chengdu 610095, China; yangang8@sohu.com (G.Y.); shihua3434@sina.com (H.S.)

**Keywords:** quantum particle technology, digital signals, radiation methods, concrete, crack detection, three-dimensional images

## Abstract

Cracking in hydraulic buried engineering can cause localized damage or complete structural failure, potentially resulting in catastrophic project outcomes. Traditional methods for detecting cracks in hydraulic concrete buried engineering are often insufficient in terms of reliability and accuracy. With the development and application of particle-based technology, it has been widely used in the field of crack detection. This research investigates the support pier of the Yingxiuwan Hydropower Plant and the lock pier of the Yuzixi Hydropower Plant. Employing principles from quantum physics, quantum particle non-destructive detection technology is introduced to identify crack locations. A three-dimensional simulation model is constructed and verified accurately through integration with CT scanning techniques. The results demonstrate that particle detection technology effectively detects cracks in hydraulic concrete buried engineering, exhibiting minimal susceptibility to external interference. The particle detection data enable 3D visualization of cracks, accurately reflecting the conditions within embedded concrete components. This method provides a reliable and advanced technical solution for precise crack detection in concrete-embedded engineering and offers critical data for exploring crack propagation mechanisms.

## 1. Introduction

In recent years, China’s water conservancy engineering construction has made remarkable achievements, and the design and safe operation of a large number of water conservancy projects indicate that China’s dam engineering technology has reached the international advanced level as a whole [1]. However, with the gradual increase in the operation time of water conservancy projects, coupled with changes in external factors and frequent occurrence of various natural disasters, a large number of concrete embedded components in water conservancy projects have experienced cracking and damage, seriously restricting project safety [2]. For example, during the safety appraisal of Longtoushan Reservoir, it was found that the dam body had multiple cracks and severe leakage [3]. Cracks parallel to the dam axis appeared in Yunzhu Reservoir, posing a great threat to the safe operation of the dam [4]. Therefore, regular safety inspections of reservoirs are essential. At the same time, China’s massive water conservancy projects have posed new challenges in detecting dangerous reservoirs [5,6].

The traditional methods for detecting structural damage in hydraulic engineering mainly include manual inspection, ultrasonic inspection [7], laser scanning inspection [8], and infrared thermal imaging inspection [9]. However, due to the high cost, subjectivity, and low efficiency of manual detection methods, their applicability in practical engineering is limited to a certain extent. In the technology of crack detection in hydraulic engineering, methods such as ultrasonic testing, laser scanning testing, and infrared imaging testing are known as non-destructive testing [10] and are highly praised by scholars for their simplicity, ease of operation, and other advantages. Yu et al. [11] used ultrasound-induced thermal imaging technology to detect surface microcracks in concrete engineering components and developed a detection system and method for collaborative excitation of multiple excitation sources, which can detect cracks with a width of 0.03 mm. Yang et al. [12] obtained an accurate method for detecting internal cracks in reinforced concrete using a deep learning approach based on a display ultrasound feature fusion neural network. Dong et al. [13] used non-destructive testing methods to conduct a reliability analysis on the fatigue of single-sided circumferential welds in offshore pipelines and risers based on existing stress intensity factor solutions. They found that the effect of misalignment on fatigue strength is not as significant as suggested by fatigue design specifications. Zou et al. [14] successfully detected voids with a depth of 150 mm inside concrete using laser-induced acoustic detection technology. Jiao et al. [15] developed a laser multi-mode scanning thermal imaging (SMLT) method that combines a fast scanning mode using a linear laser with a fine scanning mode using a point laser on the surface of the tested sample, thereby achieving surface crack detection of thermal barrier coatings (TBCs). Wang et al. [16] proposed a quantitative detection method for surface oblique cracks based on full-field scanning data. By analyzing different ultrasonic signals in the full-field scanning data from laser ultrasound, the width, angle, and length of surface-angled cracks can be determined. Shi et al. [17] proposed a non-destructive testing technique based on the Multi-Scale Enhanced Master R-CNN model by using infrared thermal imaging to detect internal cracks in thermal barrier coatings of turbine blades while considering the impact of these cracks on the surrounding environment, thereby providing more comprehensive functionality for crack detection. Kylili et al. [18] summarized the latest literature and research on passive and active infrared thermal imaging and elaborated on the basic knowledge of IRT, introducing the thermal imaging process used for building diagnosis. Zhang et al. [19] designed a universal multi-photon entangled N00N state super-resolution quantum imaging system based on the resolution advantage of N-photon entanglement quantum imaging, which significantly improved the resolution of the imaging system. Massabuau et al. [20] proposed a method for characterizing InGan/Gan quantum well interfaces through X-ray reflectivity, which is expected to be supplemented by transmission electron microscopy analysis because it is non-destructive, fast, and allows for a multi-directional characterization of roughness.

In order to observe the internal structure of cracks more intuitively, intelligent monitoring methods based on computer vision have been widely used for the detection and evaluation of dangerous engineering and have also achieved three-dimensional spatial visualization of cracks, better serving the non-destructive detection of cracks in practical engineering [21,22]. Tu et al. [23] identified the leakage channels of the Longfengshan Reservoir earth rock dam through non-destructive testing using tomographic imaging and verified the reliability of the method. Min et al. [24] combined intelligent recognition technology with the fine finite element method (XFEM) to simulate the hydraulic fracturing resistance of actual cracks in concrete dams and used image processing technology and non-destructive testing technology to identify and extract the direction and depth of actual cracks, ultimately generating a three-dimensional geometric model of the actual cracks, providing a new method for non-destructive testing. Xu et al. [25] proposed a dam crack image detection model based on dam feature enhancement and attention mechanism to address the problem of the unavailability of dam crack image samples and the low accuracy of traditional algorithms for crack detection. The model can accurately detect the location of crack targets. Khatir et al. [26] combined two optimization techniques, the extended finite element method (XFEM) and extended geometric analysis (XIGA), to propose an innovative, intelligent method based on inverse problems, which can accurately predict the location of cracks in plate structures. Agathos [27] and Chatzi [28] proposed a new numerical scheme based on the extended finite element method (XFEM) combined with a genetic algorithm, which can be applied to detect multiple cracks in three-dimensional structures. Malek et al. [29] combined computer vision algorithms with AR software and used video processing algorithms to locate fatigue cracks during the inspection process. Based on the detection results, holograms were generated and anchored at the crack positions of the inspected structure. Zhang et al. [30] proposed an automatic detection and segmentation method for surface cracks of concrete structure bridges based on computer vision deep learning models. Zhang et al. [31] proposed an intelligent model based on object detection algorithms in computer vision for road crack detection, which can accurately detect and classify four types of cracks. Qi et al. [32] proposed a three-step framework based on computer vision on the basis of traditional crack detection, which can quickly identify concrete cracks and automatically recognize their length, maximum width, and area in damage images. Zhang et al. [33] proposed a lightweight and efficient BLS-based crack detection model for automatic classification and recognition of crack images and introduced a sliding window algorithm that can efficiently identify and locate cracks in large-sized images. Therefore, it is not difficult to see that vigorously promoting the application of non-destructive testing in engineering practice can better solve the internal inspection problems of a large number of buried projects, and boldly introducing high-precision and cutting-edge detection technologies is particularly important.

This study introduces particle quantity non-destructive testing technology and examines its application using two concrete buried piers from the Yingxiuwan Hydropower Plant and the Yuzixi Hydropower Plant as research subjects. We compared the data obtained from particle-based detection with the actual crack locations for validation. The investigation focuses on advancing the technology for concrete crack detection. While ensuring the precise execution of crack detection in engineering applications, this study validates the scientific basis and reliability of particle quantity detection technology. This approach offers an effective and dependable method for accurately identifying cracking phenomena in concrete buried structures. Additionally, it provides critical data to support further exploration of crack propagation mechanisms and serves as technical support for promoting and applying particle detection technology in hydraulic engineering and related fields.

## 2. Project Overview

The Yingxiuwan and Yuzixi Hydropower Plants are runoff diversion facilities located in Wenchuan County, Sichuan Province, along the upper reaches of the Minjiang River. These plants serve as primary power transmission hubs for the Chengdu area. The Yingxiuwan Hydropower Plant commenced construction in September 1965, with its first unit generating electricity by September 1971. The project was completed in 1972, featuring an installed capacity of 140 MW and an annual power output of 713 million kW·h. Similarly, the Yuzixi Hydropower Plant connected its first unit to the grid in September 1966, achieving completion in December 1972, with an installed capacity of 160 MW and an annual output of 960 million kW·h [34].

Following nearly five decades of operation, geological stress, environmental factors, and, in particular, the “5.12” Wenchuan earthquake significantly impacted both facilities. The concrete buried components, notably the intake structures at Yingxiuwan and the support piers at Yuzixi exhibited varying degrees of cracking. A novel particle detection technology was adopted to characterize these cracks for detailed analysis.

Yingxiuwan Hydropower Plant is situated on the left bank of the Minjiang River, with an elevation ranging from 940 to 950 m. The facility experiences no overflow during the dry season. Its piers, constructed as reinforced concrete components, measure 1.5 m in width and 5 m in height. Cracking predominantly occurs on the backwater surface of the upper support piers, with the crack fronts located about 1.4 to 1.6 m below the top of the pier platform, as depicted in Figure 1 and Figure 2.

The support pier of the Yuzixi Hydropower Plant features a three-hole breast wall measuring 8 × 9 m and a bottom-hole flood sluice. It is situated in Yuzixi, a tributary on the upper right bank of the Minjiang River. Cracks have been observed in the concrete structure supporting the hinge of the sluice pier. These cracks are predominantly located on the side surfaces in the middle and upper portions of the pier. The crack fronts extend to a distance of 2.5 to 3.0 m from the top of the pier platform, as illustrated in Figure 3 and Figure 4.

## 3. Principle and Measurement of Quantum Particle Detection

### 3.1. Principle and Equipment for Measuring Particle Detection

The RSM-II measuring particle detector (Figure 5) is employed for the detection process, enabling the precise emission of quantum particles, such as photons or neutrons, into the dam structure. Utilizing the dual wave and particle properties of quantum particles, interactions between particles and structural cracks result in phase changes in the quantum particle wave function at the crack edges. These changes create a distinctive interference pattern, which influences the signal characteristics captured by the detector. By conducting a high-precision analysis of the collected quantum signals, critical parameters, including the crack’s specific location, depth, and shape, can be accurately determined [35]. The schematic diagram of the working principle of the particle detection device is shown in Figure 6.

Compared to traditional detection methods, the quantitative particle non-destructive detection technique offers superior accuracy and maintains the integrity of the dam structure during safety assessments. This advanced approach ensures that safety monitoring is conducted without inflicting damage, providing an innovative solution for health monitoring and maintenance of dams. Its application is particularly significant for the early detection of potential safety risks and for ensuring the long-term stability and reliable operation of dam structures.

### 3.2. Experiment

To ensure the scientific reliability of the detection technology, laboratory-prepared reinforced concrete marker blocks containing cracks were utilized prior to field detection, as illustrated in Figure 7. These blocks were employed to obtain standard parameters for particle detection of cracks and to establish baseline data for calibration and detection of concrete cracks. Furthermore, baseline data for the detection of hydraulic concrete cracks using particle detection technology were collected at the Qingfengling Teaching Power Station site (test subjects shown in Figure 8 below). These data were used to preliminarily assess the detection accuracy of particle detection technology for crack identification. Simultaneously, a geoelectric, electromagnetic survey instrument was employed under the same conditions to detect the cracks, enabling a comparative analysis of the detection performance between the two technologies.

Initially, the relationship between the particle detector’s response and crack characteristics was determined through controlled laboratory tests. Subsequently, repeated experiments were conducted to establish a standard correlation between crack width and the particle signal response unit value. Finally, the laboratory samples of concrete cracks were subjected to testing, generating extensive detection data. At the Qingfengling Teaching Power Station, a specific step platform was selected as the detection area, with measurement points arranged at 20 cm intervals across a grid. Both the particle detection system and the geoelectric, electromagnetic survey instrument were used to perform measurements at these locations. The measurement data are presented in Table 1. Based on the particle detection data, a 3D model of the step cracks was constructed (shown in Figure 9 and Figure 10).

The results from both laboratory-based particle detection of hydraulic concrete cracks and field measurements at the Qingfengling Teaching Power Station, using both particle detectors and geoelectric electromagnetic survey instruments, reveal that the particle detection system is capable of detecting cracks within hydraulic concrete. As long as a crack is present, the instrument responds accordingly. Moreover, the grade of the hydraulic concrete has minimal impact on the particle detection system, with little to no effect on the detection data. Additionally, there is a positive correlation between crack width and the instrument’s response value: wider cracks elicit higher response values, while narrower cracks produce lower response values. Currently, the smallest crack width detectable by the instrument is 0.5 mm, indicating that the device has a resolution of 0.5 mm.

### 3.3. Field Crack Measurement Particle Detection

Based on the apparent crack data collected at the Yingxiuwan and Yuzixi Hydropower Plants, measurements were conducted to determine the distance, width, and location of cracks relative to the top platform. Given the similarities in crack orientation and depth between the sluice piers at both plants, a consistent distribution principle was applied for measurement.

The measuring points were organized using the top of the support (gate) pier as the reference platform, with the starting point set at the top edge of the pier. Measuring points were arranged in a 10 cm grid pattern, numbered sequentially as 01, 02, and 03 from the outermost to the innermost point. A total of 18 longitudinal lines were preset, continuing until no particle signal response to cracks was detected. Horizontal lines were designated alphabetically (A, B, C, etc.) and extended until no crack-related signal response was observed. The layout of the measuring points on the support (gate) piers is depicted in Figure 11.

The trapezoidal support (gate) pier is segmented into two detection areas for data acquisition based on particle detection: the vertical detection area and the inclined detection area, separated by the top boundary of the trapezoidal pier. In the vertical detection area, grid points (e.g., A0, A1, B0, B1) are established on the top surface (aisle) of the support (gate) pier, with these points serving as base points for the particle beam wave meter. Vertical downward detection is conducted, and the collected data are labeled sequentially as A0, A1, B0, B1, and so on. In the inclined detection area, the same grid points (e.g., A0, A1, B0, B1) on the top of the pier are used as base points. Signals are emitted parallel to the inclined plane of the trapezoidal body, set at a 59° angle, to perform oblique distance detection of cracks. The data collected in this area are recorded as A0’, A1’, B0’, B1’, and so forth, as illustrated in Figure 12.

## 4. Three-Dimensional Visual Analysis of Cracks

### 4.1. Quantitative Particle Detection Data Processing

The particle detector was utilized to perform crack detection in the concrete-buried engineering of the two hydropower stations. Specifically, Pier 2# of the spillway gate at the Yuzixi Hydropower Plant and Pier 1# of the diversion dam at the Yingxiuwan Hydropower Plant were selected as examples for this analysis. Data such as crack depth and width were collected during the detection process. The collected data were subsequently organized and analyzed using particle quantity correction parameters obtained from laboratory tests on concrete cracks for the two stations. The detailed results of these analyses are presented in Table 2, Table 3, Table 4 and Table 5.

### 4.2. Three-Dimensional Fracture Model

Based on the crack detection measurement data, the three-dimensional coordinates derived from the crack geometry data were imported into CAD. The spatial points were sequentially connected to create a three-dimensional complex surface comprising multiple smaller surfaces. This constructed 3D crack surface was then imported into ANSYS, where the complex surface was unified into an independent three-dimensional surface. Using this process, a three-dimensional spatial crack visualization system was established for Pier 1# of the Yingxiuwan Hydropower Plant’s support structure and Pier 2# of the Yuzixi Hydropower Plant’s spillway structure, as illustrated in Figure 13 and Figure 14.

## 5. Field Coring Verification

### 5.1. Field Coring

To further validate the reliability of particle detection technology, core drilling was conducted at the two hydropower station sites to directly and accurately reveal the internal conditions of the concrete structures. At the Yingxiuwan Hydropower Plant, the characteristic trends of existing concrete cracks were analyzed to design the coring angle. A core sample of 1 m in length and 100 mm in diameter was drilled along the direction of crack propagation, as depicted in Figure 15a. At the Yuzixi Hydropower Plant, drilling was performed based on predetermined positions, angles, and lengths, resulting in a core sample measuring 30 cm in length and 50 mm in diameter, as shown in Figure 15b. Upon completion of core drilling, each core segment was labeled with detailed records of its specific position, orientation, and depth. Following the documentation of apparent data, the core samples were cleaned to remove dust and impurities from their surfaces. This preparation ensured that the samples were suitable for subsequent laboratory analysis and testing.

### 5.2. Three-Dimensional Crack Verification

Based on the 3D visualization results of concrete cracks, the accuracy and alignment of crack characteristic parameters detected by the quantitative particle method were preliminarily assessed. Using Pier 1# of the Yingxiuwan Hydropower Plant’s water diversion dike as an example, verification was conducted to confirm whether the identified crack coincided with its location in the three-dimensional model. According to the core drilling results, the concrete sample labeled 4# from Pier 1# measured 28.5 cm in length and 75.30 cm in depth along the drilling hole. Taking the drilling hole’s location as the origin of coordinates, with the horizontal leftward direction as the positive *X*-axis and the vertical downward direction as the positive *Y*-axis, the coordinates at the two ends of the 4# concrete sample were determined to be (49.4, 53.23) and (68.09, 74.75).

In the 3D visual crack diagram, the position of concrete sample 4# was plotted with the drilling hole as the origin, as shown in Figure 16 and Figure 17. The analysis revealed that the on-site drill core location intersected with the 3D visual model of the crack surface on Pier 1# of the Yingxiuwan Hydropower Plant. This finding confirms that the crack parameters determined through particle detection are rational and that the 3D visualization model exhibits a high degree of alignment with actual conditions.

## 6. CT Scan Verification

### 6.1. CT Scan of Cracks in Concrete

Based on the in situ core drilling results, concrete samples with both visible surface cracks and no apparent cracks were selected for laboratory CT scanning (Figure 18) to investigate the internal expansion patterns of concrete cracks. The reconstructed CT crack model was systematically compared with the previously established 3D visualization model and the in situ core drilling results (Figure 19 and Figure 20). Key parameters such as crack location and width were analyzed across these models. The comparisons confirmed a strong correlation between the data sets, validating the reliability of the particle detection technology in accurately identifying and characterizing cracks within concrete-buried engineering structures.

Representative concrete samples from the Yingxiuwan and Yuzixi Hydropower Plants were subjected to CT scanning and reconstruction to analyze crack characteristics. During data processing, a range of crack parameters were selected as output data, measured in millimeters. These included probability, diameter, volume, voxel, surface, edge distance, and gap, as well as positional coordinates Pos X, Pos Y, and Pos Z.

### 6.2. Verification of Fracture CT Reconstruction Model

Using Pier 2# of the spillway gate at Yuzixi Hydropower Plant as an example, Figure 21 illustrates the angle and depth of in situ drilling used to establish a 3D model of the drilling and coring process. The red section in Figure 22 represents the channel formed by the in situ drilling. For the corresponding CT-scanned drill core sample, the intersection plane between the sample and the detected crack was precisely identified and marked in red within the 3D visualization model, as shown in Figure 23. The CT scanning reconstruction of the fracture profile revealed a cross-sectional shape that aligned closely with the red-highlighted area in the 3D model. This strong correlation further confirms the high accuracy of the quantitative particle detection method in identifying and characterizing concrete cracks.

Additionally, as crack depth increases, the detection results maintain consistency without significant deviation, demonstrating the high accuracy of the method. This level of precision fully satisfies the requirements for detecting crack depths and evaluating the structural safety of concrete-buried engineering structures. The application of particle detection technology in crack detection for concrete buried engineering has proven to be both highly accurate and reliable, making it a robust tool for structural assessment and maintenance.

## 7. Conclusions

Conventional non-destructive testing methods for crack detection in concrete buried engineering are often limited by factors such as site conditions and detection depth. To address these challenges, a novel particle detection technology has been developed, offering reduced susceptibility to external interference improved depth penetration, and suitability for operation in confined spaces. Field detection and 3D visual modeling of cracks in the embedded concrete components of the Yingxiuwan and Yuzixi Hydropower Stations demonstrated the feasibility, reliability, and accuracy of this technology. These findings were further corroborated by in situ core sampling and laboratory CT scanning reconstruction. This study provides a promising approach for crack detection in concrete-embedded structures in hydraulic engineering, with significant scientific relevance and potential for widespread application.

## Figures and Tables

**Figure 1 sensors-25-00683-f001:**
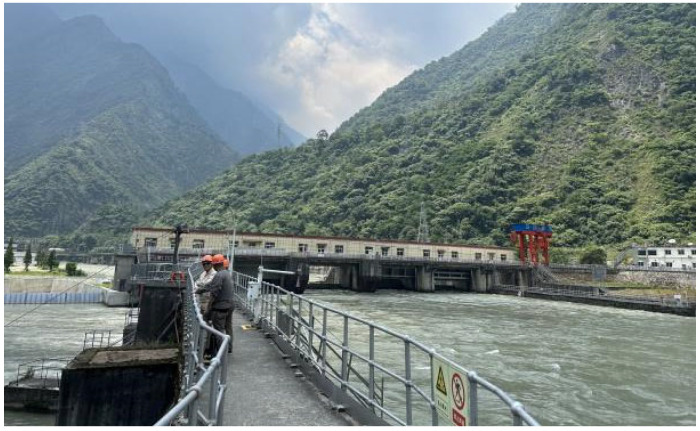
Situation of the head of the Yingxiuwan Hydropower Plant.

**Figure 2 sensors-25-00683-f002:**
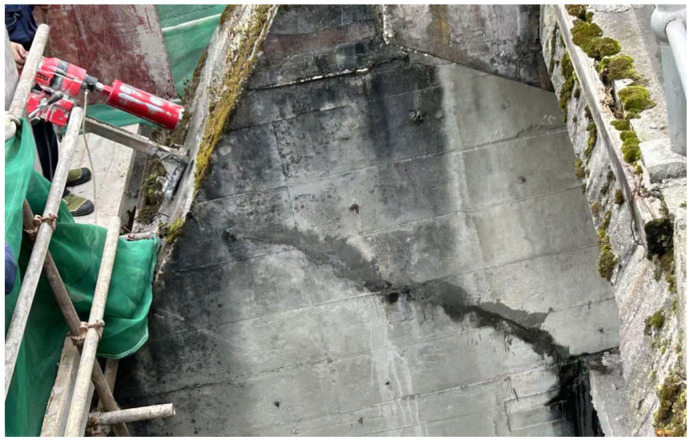
Cracks in buttress piers of Yingxiuwan Hydropower Plant.

**Figure 3 sensors-25-00683-f003:**
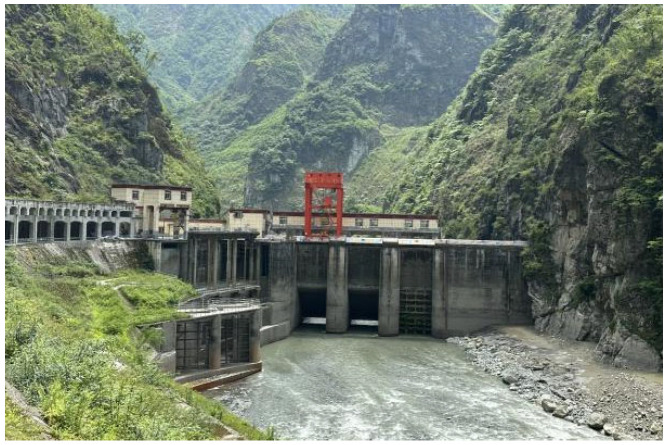
Situation of the head of the Yuzixi Hydropower Plant.

**Figure 4 sensors-25-00683-f004:**
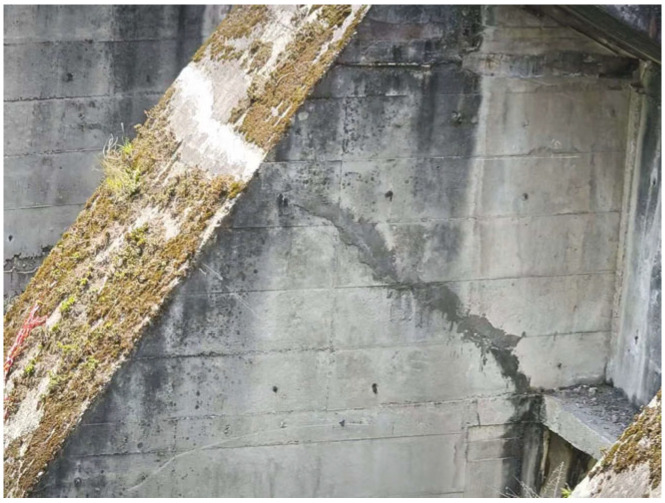
Cracks in buttress piers of Yuzixi Hydropower Plant.

**Figure 5 sensors-25-00683-f005:**
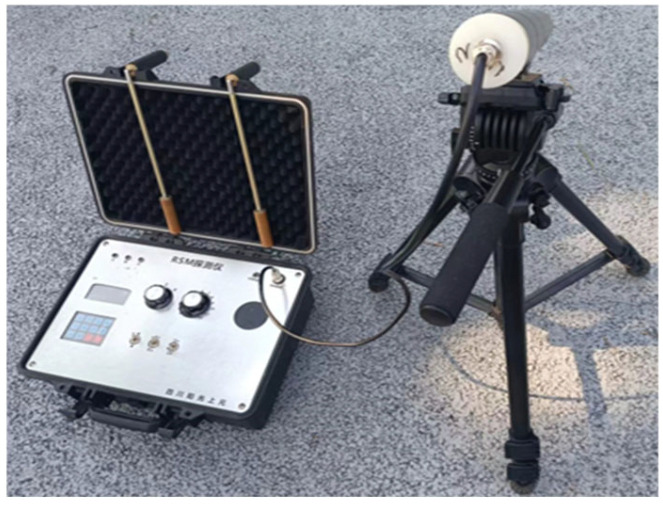
Measuring particle detector.

**Figure 6 sensors-25-00683-f006:**
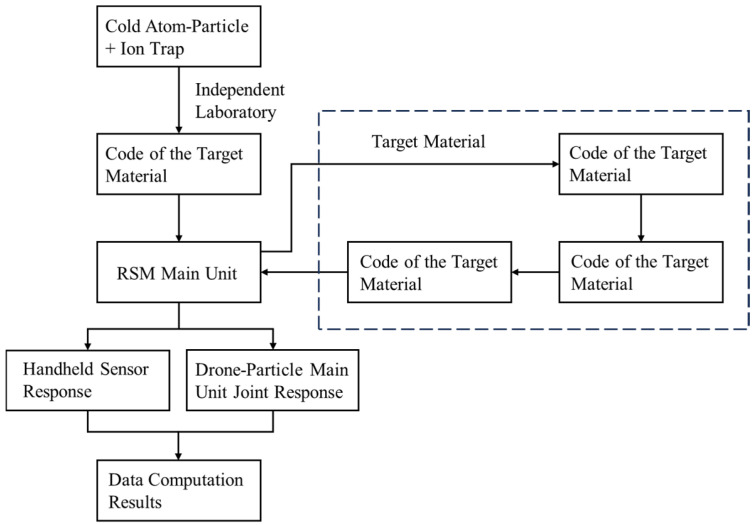
The working principle diagram of the particle detection device.

**Figure 7 sensors-25-00683-f007:**
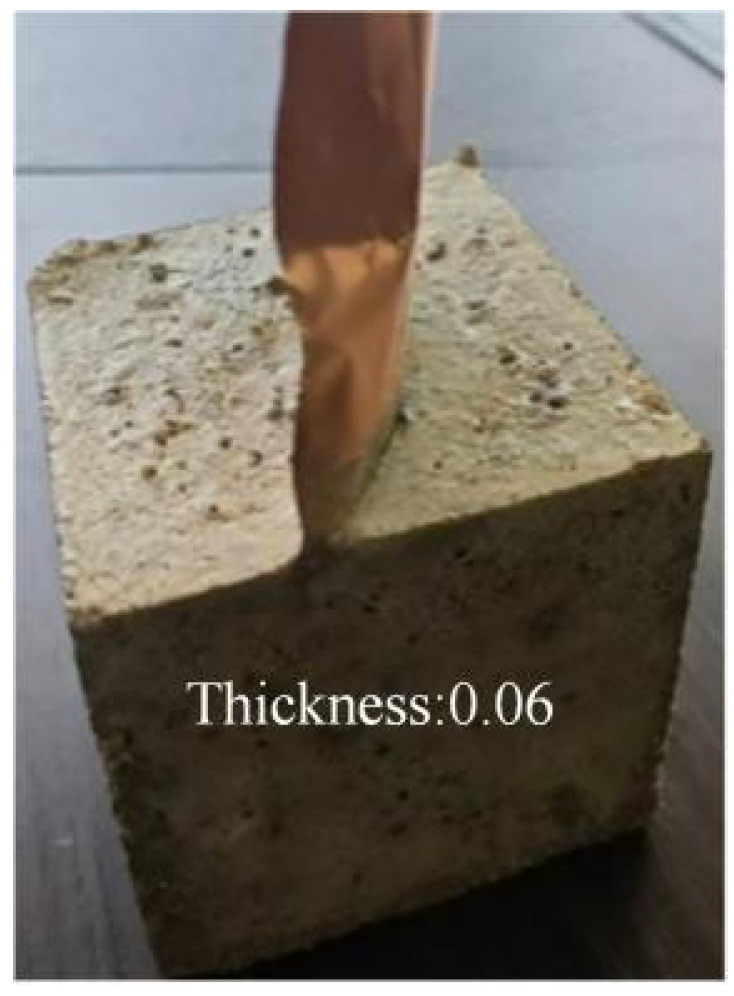
Crack concrete sample block.

**Figure 8 sensors-25-00683-f008:**
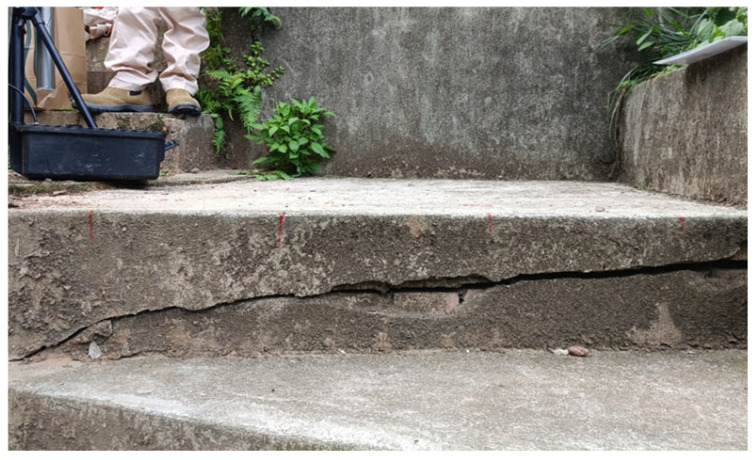
Crack diagram of the staircase.

**Figure 9 sensors-25-00683-f009:**
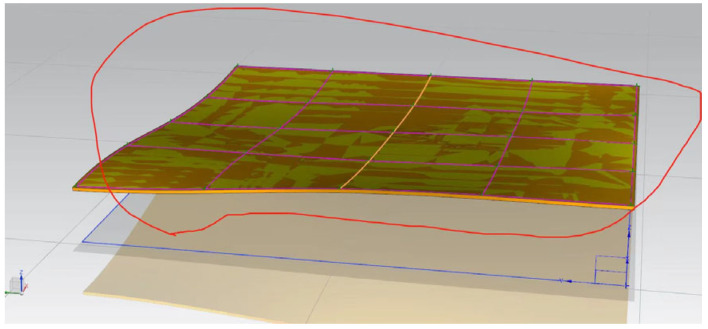
Three-dimensional model of step cracks.

**Figure 10 sensors-25-00683-f010:**
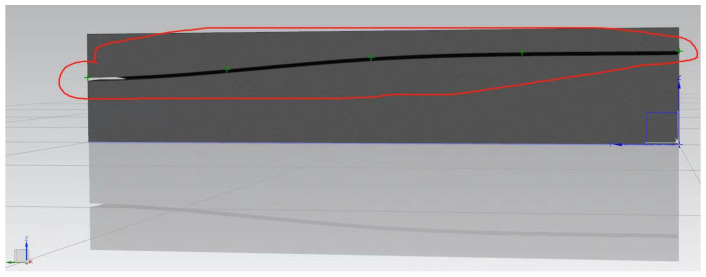
Cross-sectional variation in the crack.

**Figure 11 sensors-25-00683-f011:**
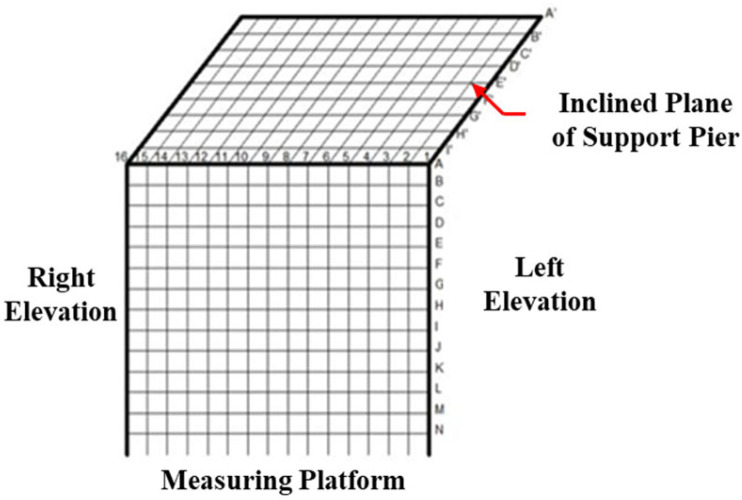
Support pier measurement layout diagram.

**Figure 12 sensors-25-00683-f012:**
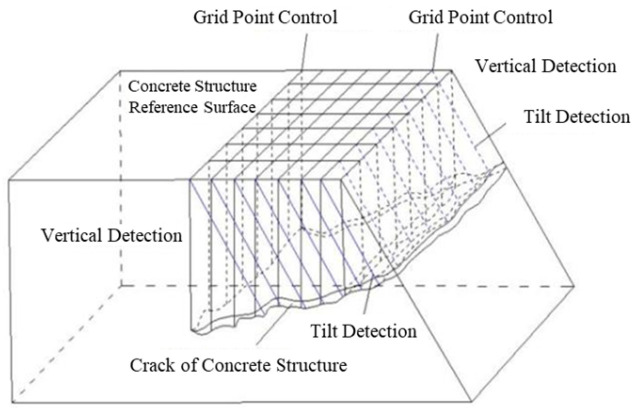
Schematic diagram of measuring and distributing the pier.

**Figure 13 sensors-25-00683-f013:**
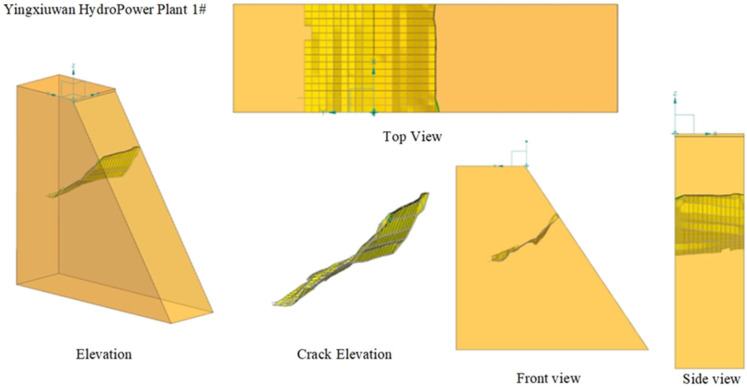
A three-dimensional visualization of the crack in Pier 1# of the Yingxiuwan Hydropower Plant’s support structure.

**Figure 14 sensors-25-00683-f014:**
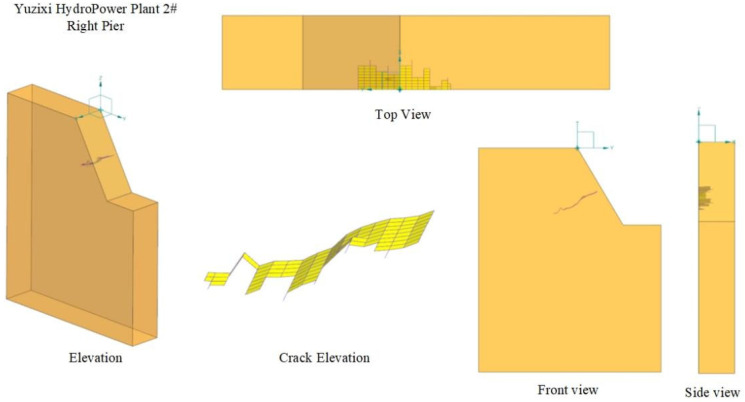
A three-dimensional visualization of the crack in Pier 2# of the Yuzixi Hydropower Plant’s spillway structure.

**Figure 15 sensors-25-00683-f015:**
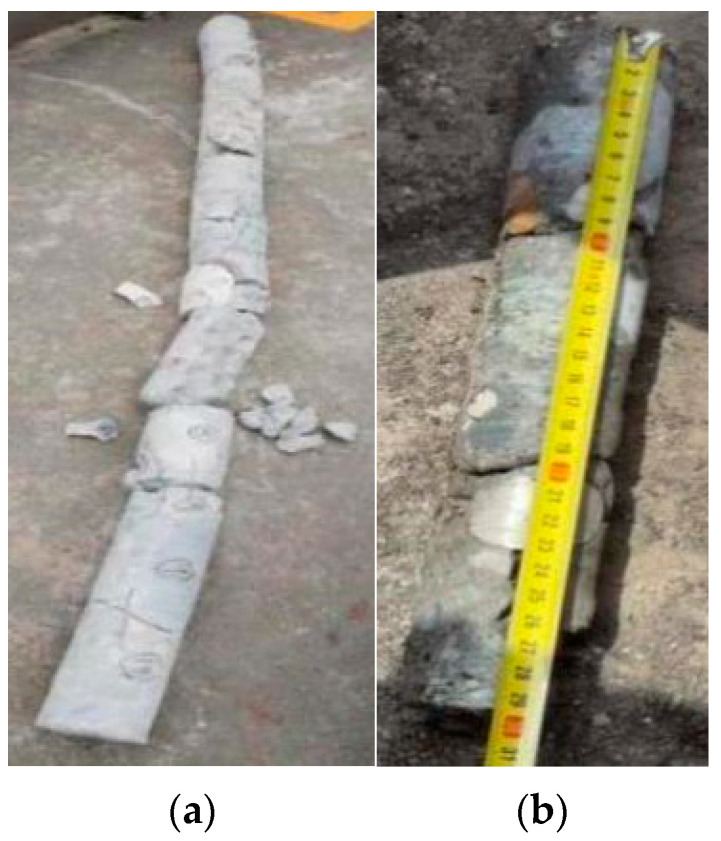
Core sample from Hydropower Plant (**a**) Yingxiuwan and (**b**) Yuzixi.

**Figure 16 sensors-25-00683-f016:**
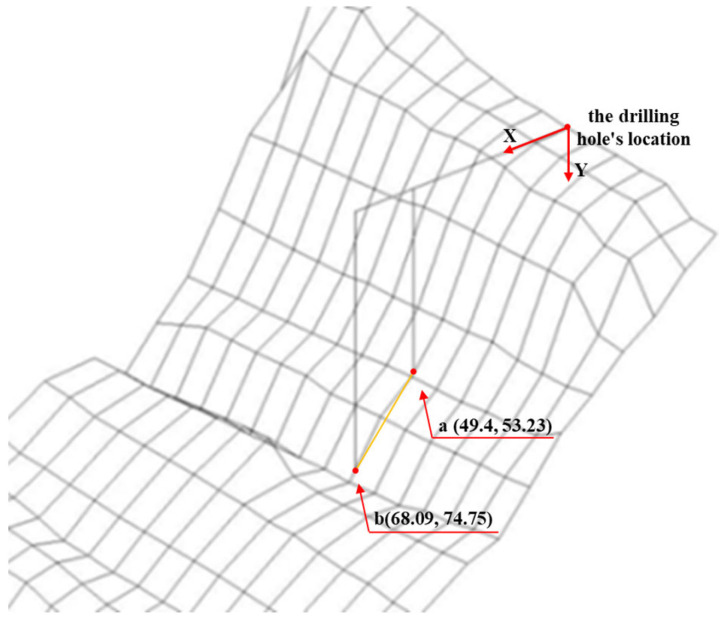
Yingxiuwan Hydropower Plant: Pier 1# crack verification elevation.

**Figure 17 sensors-25-00683-f017:**
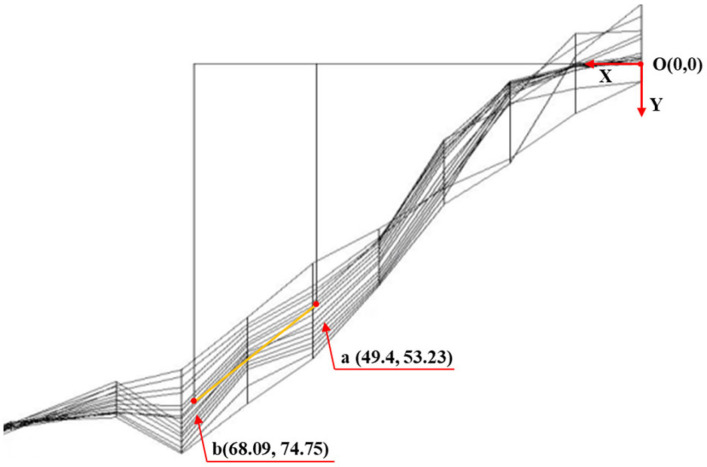
Side view of crack verification of Pier 1# (buttress pier) at Yingxiuwan Hydropower Plant.

**Figure 18 sensors-25-00683-f018:**
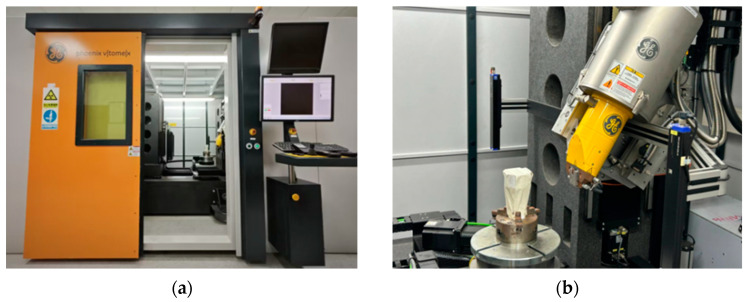
Laboratory CT scanner (**a**) operating platform and (**b**) CT scanner.

**Figure 19 sensors-25-00683-f019:**
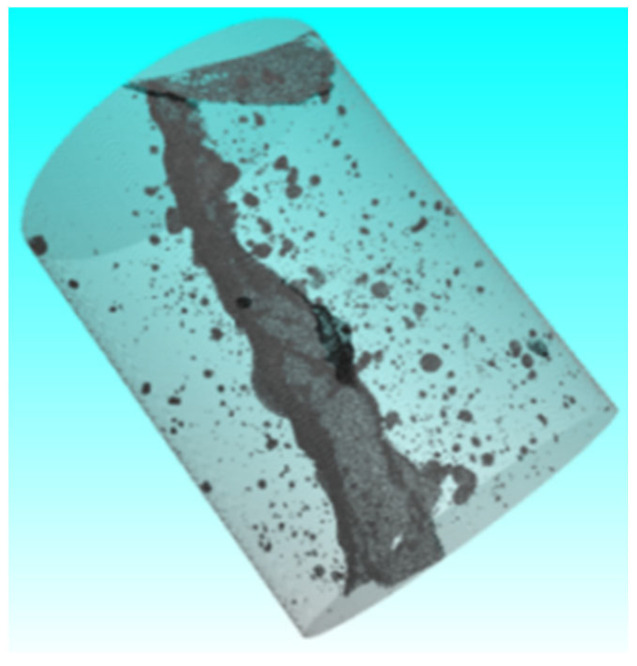
Concrete crack reconstruction model of Pier 1# (buttress pier) of Yingxiuwan Hydropower Plant.

**Figure 20 sensors-25-00683-f020:**
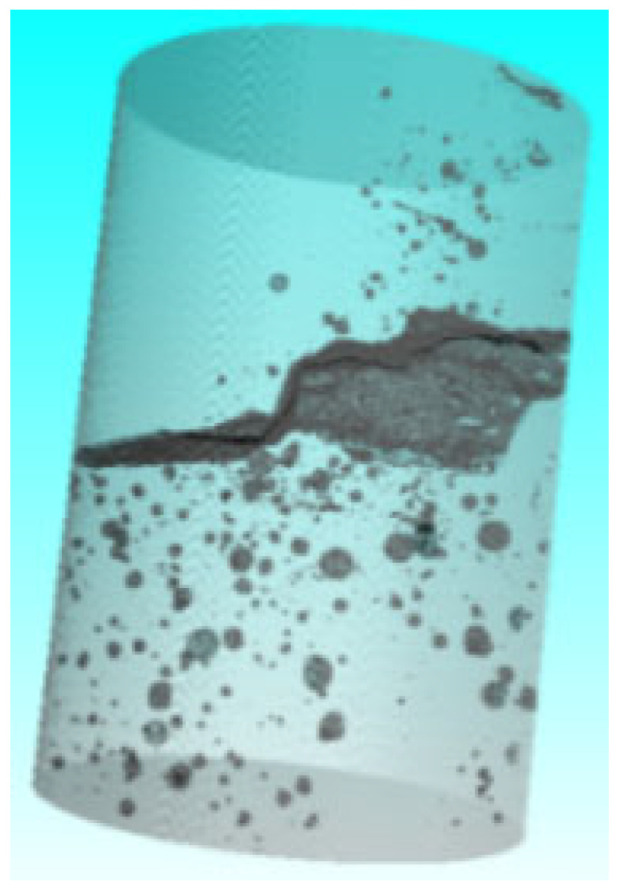
Reconstruction model of the concrete crack of 2# sluice pier in Yuzixi Hydropower Plant.

**Figure 21 sensors-25-00683-f021:**
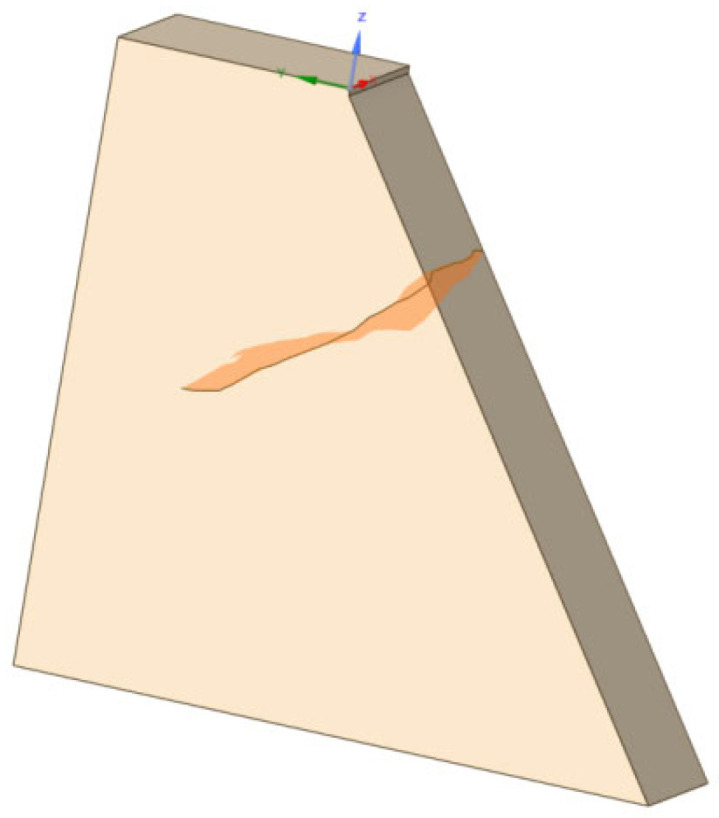
Three-dimensional visual model of crack in Pier 2# of Yuzixi Hydropower Plant’s spillway structure.

**Figure 22 sensors-25-00683-f022:**
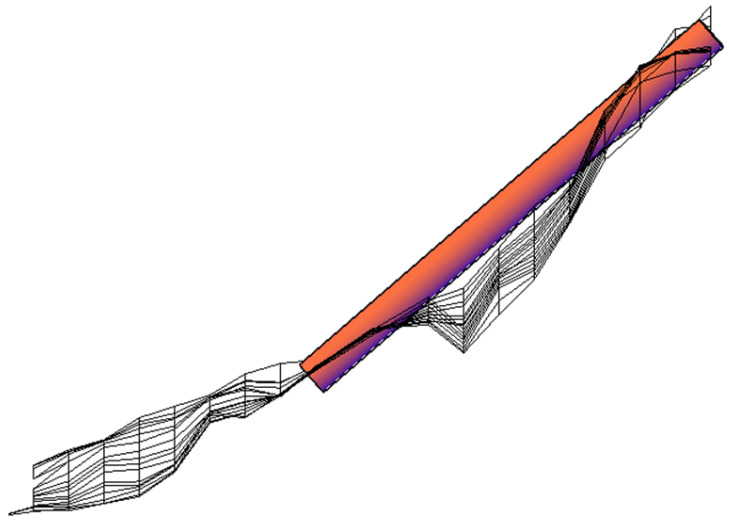
Schematic diagram of crack core drilling in Pier 2# of Yuzixi Hydropower Plant’s spillway structure.

**Figure 23 sensors-25-00683-f023:**
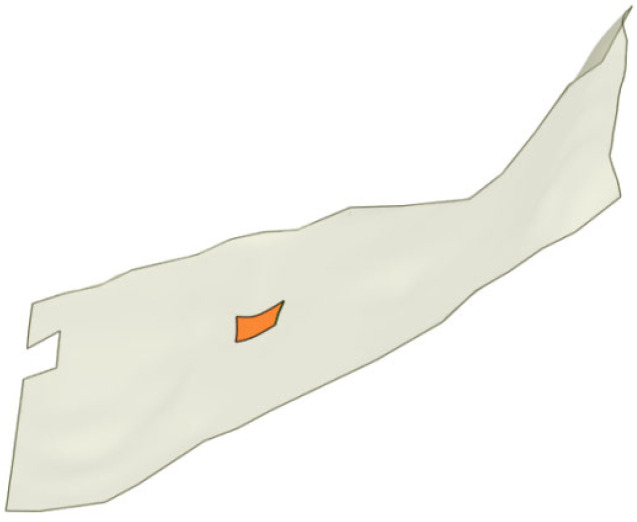
Crack intersection model of lock pier core drilling test in Pier #2 at Yuzixi Hydropower Plant.

**Table 1 sensors-25-00683-t001:** Survey data for Pier 1# branch of diversion levee at Yingxiuwan Hydropower Plant.

Measurement Point ID	Detection Depth (m)	Detection Direction (°)	Instrument Response Depth (m)	Response Magnitude
1	5	90	0.5~0.7	20
2	5	90	0.5~1	20
3	5	90	0.5~1	23
4	5	90	0.5~1	23
5	5	90	0.5~1	25

**Table 2 sensors-25-00683-t002:** The crack measurement particle detection data for Pier 1# (buttress pier) at the Yingxiuwan Hydropower Plant.

Projects	Penetrability	Fracture Length (m)		Crack Width (mm)		Maximum Crack Depth (m)
		Measured Length	Penetration Length	Front	End	
1# Buttress left side	Penetrate	2.03	1.52	10–15	25	1.5
1# Buttress right side		2.01	1.4	10–15	3–8	1.5

**Table 3 sensors-25-00683-t003:** Survey data for Pier 1# branch of diversion levee at Yingxiuwan Hydropower Plant.

Number		1	2	3	4	5	6
A’	Depth (m)	1.36	1.36	1.305	1.31	1.31	1.315
	Slit width (mm)	15	12	12	12	12	12
B’	Depth (m)	1.43	1.375	1.32	1.327	1.324	1.325
	Slit width (mm)	13	14	12	16	15	15
C’	Depth (m)	1.53	1.407	1.412	1.364	1.371	1.382
	Slit width (mm)	15	15	13	15	14	14

**Table 4 sensors-25-00683-t004:** Detection data for Pier 2# of spillway gate at Yuzixi Hydropower Plant.

Projects	Penetrability	Fracture Length (m)		Crack Width (mm)		Maximum Crack Depth (m)
		Measured Length	Penetration Length	Front	End	
1# Buttress left side	Unpenetrated	2.92	2.675	1.03–1.45	0.11–0.24	0.9
1# Buttress right side		3.36	3.30	1.1–2.0	0.6–0.9	1

**Table 5 sensors-25-00683-t005:** Crack distribution detection data for Pier 2# of spillway gate at Yuzixi Hydropower Plant.

Number		1	2	3	4	5	6
A’	Depth (m)	2.992	2.992	2.990	2.986	2.984	2.980
	Slit width (mm)	1.40	1.40	1.44	1.42	1.42	1.40
C’	Depth (m)	3.130	3.128	3.126	3.124	3.120	3.120
	Slit width (mm)	1.32	1.30	1.30	1.30	1.30	1.28
E’	Depth (m)	3.241	3.242	3.240	3.237	3.237	3.234
	Slit width (mm)	1.02	1.02	1.00	1.00	1.00	1.00

## Data Availability

Data are contained within the article.

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
