# Peer review of "Research on Concrete Crack Detection in Hydropower Station Burial Engineering Based on Quantum Particle Technology"

_sensors, 2025, doi:10.3390/s25030683_

Round 1

Reviewer 1 Report

Comments and Suggestions for Authors

In this manuscript, the concrete crack detection method in hydropower station 2 Burial engineering was studied based on quantum particl technology.  The study method is feasible and is of great significance for hydrodynamic devices. Before the paper is accepted, the following revisions are recommended.

1.     The introduction part should construct a more comprehensive literature review framework. Add some literature on non-contact crack measurements appropriately.

For example: Licheng S ,Yun L ,Yuzhang W , et al.Evaluation of Internal Cracks in Turbine Blade Thermal Barrier Coating Using Enhanced Multi-Scale Faster R-CNN Model[J].Applied Sciences,2022,12(13):6446-6446.

2.     It is recommended to rearrange the format of pictures and references in the text according to the requirements of the journal.

3.     Some pictures should be further modified to make it easier for readers to understand and accept.

4.     The sentences in the abstract should be further revised to make the article more logical.

5.     The validation part is not reflected in Figure 13 and Figure 14. Please improve it.

Comments on the Quality of English Language

English speaking researchers to help improve English writing.

Author Response

Comments 1: The introduction part should construct a more comprehensive literature review framework. Add some literature on non-contact crack measurements appropriately.
Response 1: Thank you for your valuable suggestions. In the revised manuscript, we have divided the literature review in the introduction into two main sections. In the first section, we discuss the current major traditional methods for structural health monitoring. In this part, we have added some recent references to enhance the existing review on non-contact crack detection technologies, such as Licheng Shi et al., 2021; Licheng Shi et al., 2022.The second section focuses on computer vision-based intelligent detection methods, and we have included additional references, such as Zhang et al., 2022; Qi et al., 2024, to highlight the significance of these methods for structural health monitoring.
The changes can be found in the second and third paragraphs of the introduction in the revised manuscript.
Comments 2: It is recommended to rearrange the format of pictures and references in the text according to the requirements of the journal.
Response 2: Thank you very much for your suggestion. We have conducted a thorough review of the formatting for all figures and tables and have made revisions in accordance with the journal’s requirements. Similarly, the reference list has been updated to conform to the journal's citation style, and we have also included DOI links (https://doi.org/) to facilitate easy access to the referenced articles for interested readers.
Comments 3: Some pictures should be further modified to make it easier for readers to understand and accept.
Response 3: Thank you for your valuable reminder. We have carefully reviewed the images throughout the manuscript. We have replaced Figures 2 and 4 with higher-resolution images of the crack damage, taken from a closer distance, to better display the cracks on the supporting piers of the hydropower station. Figure 7 had poor quality, with many symbols being difficult to read, so we have processed all graphical elements to improve clarity. Figures 13 and 14, originally validation images of the cracks on the supporting piers, have been updated to include a 3D coordinate system for better presentation. The sentences in the abstract should be further revised to make the article more logical.
Comments 4: The sentences in the abstract should be further revised to make the article more logical.
Response 4: Thank you for your valuable suggestions. In the abstract, we have highlighted the significant advancements in quantum particle technology and introduced the application of quantum particle detection techniques in our study. Additionally, we have optimized the phrasing in the abstract to improve clarity and coherence.
Comments 5: The validation part is not reflected in Figure 13 and Figure 14. Please improve it.
Response 5: Thank you for your suggestion. We have re-optimized Figures 13 and 14 and included coordinate systems and borehole channels in the images. This will help readers more clearly compare the actual crack locations with the crack positions detected by the particle detection method.
The changes can be found in Figures 12 and 13 of the revised manuscript.

Reviewer 2 Report

Comments and Suggestions for Authors

SUMMARY

The article submitted for review is relevant to modern science. This research investigates the support pier of the Yingxiuwan Hydropower Plant and the lock pier of the Yuzixi Hydropower Plant using quantum physics principles. A non-destructive quantum particle detection technology is considered to determine crack locations.

The results show that the particle detection technology effectively detects cracks in underground hydraulic concrete structures, demonstrating minimal susceptibility to external interference.

The authors demonstrated a reliable and advanced technical solution for accurate crack detection in concrete-embedded structures, and therefore this study has scientific value and engineering application significance.

The reviewer believes that this article can be considered for publication, but the comments that the reviewer has are needed to be corrected. The comments are listed below.

COMMENTS

1. The authors are encouraged to structure the article into sections: Introduction – Materials and Methods – Results – Discussion – Conclusion

2. At the end of the abstract, the authors state that the technique accurately captures the conditions inside embedded concrete components. How can this accuracy be quantified? This should be added both to the abstract and to the results, where the authors state that the use of particle detection technology in crack detection for concrete underground engineering structures has proven to be highly accurate.

3. The authors should consider strengthening the Introduction section, perhaps adding more reviews of similar studies. More traditional methods of structural health monitoring can be analyzed and shortcomings or deficiencies in previous studies can be described in detail. This section should also focus on emerging intelligent monitoring methods based on computer vision.

4. At the end of the Introduction section, the scientific problem, scientific novelty, purpose and tasks of the study, as well as theoretical and practical significance should be more clearly stated.

5. Instead of Figures 2 and 4, which show cracks in the hydroelectric power station supports, it is recommended to select images (or add) where the damage is at a closer distance.

6. The main characteristics of the equipment used (RSM-II particle detector, computer tomograph) and the software version (ANSYS) should be described.

7. The article might have benefited from the addition of a program of experimental studies.

8. A description of the physical and mechanical properties of the reinforced concrete marker blocks shown in Figure 6 should be added. As well as the characteristics of the cracks in them.

9. The authors should strengthen the article by adding formulas for the processes described in the work.

10. Unfortunately, Figure 7 is of poor quality. The authors need to work with all the graphic elements, many of them have unreadable symbols.

11. A detailed comparison of the obtained results with other methods is needed, which should be done in the Discussion section.

12. The "Conclusions" section does not meet the requirements of the journal. "Conclusions" should clearly formulate the scientific and applied result, as well as the scientific and applied prospects for the application and use of these results.

13. The article provides a total of 18 references. This is very little. The list of references should be increased to at least 35-40 titles.

This article has some promise, but there is still a lot of work to be done to improve it. Overall conclusion – Major Revisions.

Author Response

Comments 1: The authors are encouraged to structure the article into sections: Introduction – Materials and Methods – Results – Discussion – Conclusion

Response 1: Thank you for your valuable suggestion. This study is based on a practical engineering case, and therefore, the background of the project is discussed in the first section of the manuscript. The second section describes the principles of particle-based detection and the experimental work conducted prior to the measurements. In the third section, the data obtained through particle-based detection is used to perform a three-dimensional reconstruction of the crack in the support pier, and the detection data is preliminarily validated against the location of the on-site boreholes. Sections four and five provide validation of the cracks through core sampling and CT scanning, respectively, to verify the feasibility of particle-based detection. The final section presents the conclusion.

Comments 2: At the end of the abstract, the authors state that the technique accurately captures the conditions inside embedded concrete components. How can this accuracy be quantified? This should be added both to the abstract and to the results, where the authors state that the use of particle detection technology in crack detection for concrete underground engineering structures has proven to be highly accurate.

Response 2: Thank you very much for your suggestion. This study is based on a real engineering case, with the aim of validating the reliability of particle-based detection technology in assessing deteriorated structures. However, there are very few examples of this technology being applied to damage detection in deteriorating structures, and currently, there are no established standards in the industry to quantitatively assess the accuracy of its detection results. Through this engineering case, we aim to validate and promote the use of this technology. We have successfully applied for a patent titled "Concrete Structure Crack Detection Method Based on Quantum Measurement Technology” Invention Patent."

Comments 3: The authors should consider strengthening the Introduction section, perhaps adding more reviews of similar studies. More traditional methods of structural health monitoring can be analyzed and shortcomings or deficiencies in previous studies can be described in detail. This section should also focus on emerging intelligent monitoring methods based on computer vision.

Response 3: Thank you for your valuable suggestions. In the revised manuscript, we have divided the literature review in the introduction into two main sections. In the first section, we discuss the current major traditional methods for structural health monitoring. In this part, we have added some recent references to enhance the existing review on non-contact crack detection technologies, such as Licheng Shi et al., 2021; Licheng Shi et al., 2022.The second section focuses on computer vision-based intelligent detection methods, and we have included additional references, such as Zhang et al., 2022; Qi et al., 2024, to highlight the significance of these methods for structural health monitoring.
The changes can be found in the second and third paragraphs of the introduction in the revised manuscript.

Comments 4: At the end of the Introduction section, the scientific problem, scientific novelty, purpose and tasks of the study, as well as theoretical and practical significance should be more clearly stated.
Response 4: Thank you for your suggestion. At the end of the introduction section, we have added a description of the research subject, research objectives, research methods, research innovations, and the significance of the study.

Comments 5: Instead of Figures 2 and 4, which show cracks in the hydroelectric power station supports, it is recommended to select images (or add) where the damage is at a closer distance.

Response 5: Thank you very much for your suggestion. In the resubmitted manuscript, we have replaced Figures 2 and 4 with close-up images of the cracks in the support pier for better demonstration.

Comments 6: The main characteristics of the equipment used (RSM-II particle detector, computer tomograph) and the software version (ANSYS) should be described.

Response 6: Thank you very much for your suggestion. In the second section, we have added the schematic diagram of the working principle of the RSM-II quantum detection device. Since this paper aims to verify the reliability of particle detection technology in the application of hazardous engineering, we have not provided extensive descriptions of the principles behind the CT scanner and ANSYS software.

Comments 7: The article might have benefited from the addition of a program of experimental studies.

Response 7: Thank you for your suggestions. In the revised version, Section 2.2 has been updated to include an additional experiment aimed at preliminarily assessing the suitability of quantum particle detection technology for crack detection. Preliminary tests were conducted on concrete containing cracks at the Qingfengling Teaching Power Station, and the minimum resolution of the quantum detection instrument was also evaluated.
The changes can be found in Section 2.2 of the summary.

Comments 8: A description of the physical and mechanical properties of the reinforced concrete marker blocks shown in Figure 6 should be added. As well as the characteristics of the cracks in them.

Response 8: Thank you for your valuable suggestions. In the revised manuscript, we have added an experiment that demonstrates the capability of the quantum particle detector to detect cracks in hydraulic concrete. The instrument responds whenever a crack is present, and the concrete grade has little effect on the instrument's performance. We assessed the depth of the cracks based on the instrument's response depth and response magnitude, which were then used to construct a three-dimensional model.

Comments 9: The authors should strengthen the article by adding formulas for the processes described in the work.

Response 9: Thank you very much for your valuable suggestions. This study is based on a real engineering case and supplemented with laboratory experiments to verify the accuracy of quantum particle detection technology in identifying certain structural damages in hydraulic engineering. As a result, the manuscript provides limited descriptions of the underlying principles and calculation processes, with a greater focus on text and graphical representations.

Comments 10: Unfortunately, Figure 7 is of poor quality. The authors need to work with all the graphic elements, many of them have unreadable symbols.

Response 10: Thank you very much for your reminder. In the resubmitted manuscript, we have revised Figure 7 to make it clearer and easier for readers to interpret.

Comments 11: A detailed comparison of the obtained results with other methods is needed, which should be done in the Discussion section.

Response 11: Thank you very much for your valuable suggestion. In this study, a three-dimensional model of the crack was constructed based on the structural data obtained from quantum particle detection. Then, concrete core samples extracted from boreholes were subjected to CT scanning for crack reconstruction, and the results were compared and validated against the quantum particle detection data. This approach was used to assess the feasibility of using quantum particle detection technology for damage detection in concrete structures within hydraulic engineering.

Comments 12: The "Conclusions" section does not meet the requirements of the journal. "Conclusions" should clearly formulate the scientific and applied result, as well as the scientific and applied prospects for the application and use of these results.

Response 12: Thank you very much for your valuable suggestion. In the conclusion section, we introduce the advantages of quantum particle detection technology compared to traditional non-destructive testing methods, leading into the research presented in this paper. We then compare and validate the three-dimensional crack model constructed based on quantum particle detection with the crack reconstruction model obtained through CT scanning. Finally, we summarize the feasibility of using quantum particle detection technology for crack detection in hydraulic engineering and highlight its high potential for future applications.

Comments 13: The article provides a total of 18 references. This is very little. The list of references should be increased to at least 35-40 titles.

Response 13: Thank you very much for your suggestion. In the revised manuscript, we have expanded the introduction section to elaborate on traditional structural damage detection methods as well as current computer vision-based intelligent detection methods. Additionally, we have included more references, bringing the total to 36.

Reviewer 3 Report

Comments and Suggestions for Authors

I recommend your article for publication with reconsider after major revisions (substantial revisions to text or experimental methods needed).
My main comments relate mainly to the technical design of the text. “Sensors” magazine has a very simple and very convenient universal template for designing articles https://www.mdpi.com/files/word-templates/sensors-template.dot
Notes:
1. I would also include the following keywords in the list: “digital signals”, “radiation methods”, “three-dimensional images”.
2. Please note the rule of the journal “Sensors” regarding section and subsection headings (dots after the number), figures and tables.
3. References should be formatted in the same way. References should be formatted in MDPI style. A link to https://doi.org/... facilitates access to the cited article for the curious reader.
Wang, S.; Xia, X.; Ye, L.; Yang, B. Automatic Detection and Classification of Steel Surface Defect Using Deep Convolutional Neural Networks. Metals 2021, 11, 388. https://doi.org/10.3390/met11030388
4. Usually, I do not insist on following some standard structure of scientific publications. But for this manuscript, including the Materials and Methods section in the text of the manuscript would have weakened the intensity of my indignation (and perhaps not only mine), see points A and B. The reader needs to understand what specific particles are being discussed, whether they are emitted by an external source (there is none in the text of the manuscript) or whether concrete is such a source. If concrete is such a source, then it is necessary to provide examples of natural radioactivity of concrete in the Materials subsection. It is necessary to include in the Methods subsection the actual methods and algorithms for forming a set of digital signals necessary to obtain spatial information on the structural state of concrete in the area of interest. If we are talking about the emission radiation method, then the reader is interested in knowing the mathematical model of the disturbance of the radiation field created by a crack of arbitrary shape by an integral concrete test object. The authors should answer the question: how the recorded radiation field emitted by the area of interest, in which there is a crack, is transformed into an estimate of the spatial shape of the crack. The manuscript lacks information on the detector characteristics. Are complex scanning trajectories possible with changes in the spatial orientation of the detector axis relative to the surface of the concrete object being tested? How does the method proposed by the authors compare with gamma and neutron methods in nuclear geophysics? This question pertains to point A, see above.
Please be careful when editing the text of the article, maybe I missed something else.

Author Response

Comments 1: I would also include the following keywords in the list: “digital signals”, “radiation methods”, “three-dimensional images”.

Response 1: Thank you very much for your suggestion. In the revised manuscript, we have made the following changes to the keywords section: we have removed "Hydropower plant" and "Burial engineering," and added "digital signals," "radiation methods," and "three-dimensional images."

Comments 2: Please note the rule of the journal “Sensors” regarding section and subsection headings (dots after the number), figures and tables.

Response 2: Thank you very much for your suggestion. In the resubmitted manuscript, we have reviewed the formatting of all figures, tables, and section titles, and have made the necessary revisions to strictly adhere to the journal's requirements.

Comments 3: References should be formatted in the same way. References should be formatted in MDPI style. A link to https://doi.org/... facilitates access to the cited article for the curious reader.

Response 3: Thank you very much for your suggestion. We have conducted a thorough review of the formatting for all references in the manuscript and have made corrections according to the journal’s requirements. Additionally, we have included DOI links (https://doi.org/) to facilitate access for interested readers to the cited articles.

Comments 4: Usually, I do not insist on following some standard structure of scientific publications. But for this manuscript, including the Materials and Methods section in the text of the manuscript would have weakened the intensity of my indignation (and perhaps not only mine), see points A and B. The reader needs to understand what specific particles are being discussed, whether they are emitted by an external source (there is none in the text of the manuscript) or whether concrete is such a source. If concrete is such a source, then it is necessary to provide examples of natural radioactivity of concrete in the Materials subsection. It is necessary to include in the Methods subsection the actual methods and algorithms for forming a set of digital signals necessary to obtain spatial information on the structural state of concrete in the area of interest. If we are talking about the emission radiation method, then the reader is interested in knowing the mathematical model of the disturbance of the radiation field created by a crack of arbitrary shape by an integral concrete test object. The authors should answer the question: how the recorded radiation field emitted by the area of interest, in which there is a crack, is transformed into an estimate of the spatial shape of the crack. The manuscript lacks information on the detector characteristics. Are complex scanning trajectories possible with changes in the spatial orientation of the detector axis relative to the surface of the concrete object being tested? How does the method proposed by the authors compare with gamma and neutron methods in nuclear geophysics? This question pertains to point A, see above.

Response 4: Thank you very much for your valuable suggestions. This study is based on a real engineering case, with the engineering background described in the first section of the manuscript. The second section elaborates on the principles of quantum particle detection and the experimental work conducted prior to measurements. In the third section, we used the data obtained from quantum particle detection to perform a three-dimensional reconstruction of the cracks in the pier and initially validated the detection data by comparing it with the actual drilling locations. Sections four and five focus on field core sampling and CT scanning to further validate the feasibility of quantum particle detection. The final section provides a summary of the findings.Additionally, in the second section, we have included a schematic diagram of the working principle of the RSM-â…¡ quantum detection equipment. In the subsection 2.2, we have added an experiment where concrete containing cracks from the Qingfengling Teaching Power Station was tested to preliminarily assess the suitability of quantum particle detection technology for crack detection.

Round 2

Reviewer 2 Report

Comments and Suggestions for Authors

The authors have revised the manuscript, taking into account most of the reviewer's comments. The article is now more suitable for publication in the journal. The reviewer has no more significant comments.

Reviewer 3 Report

Comments and Suggestions for Authors

Overall, I am satisfied with the responses to my comments.
I will propose to publish the manuscript under review in the proposed form.
However, I remain of the same opinion: a mathematical model for the formation of digital radiation signals would certainly enhance the article and make it even more interesting and significant from a scientific point of view. But this is just my point of view.